# Regulation of MMP-2 by IL-8 in Vascular Endothelial Cells: Probable Mechanism for Endothelial Dysfunction in Women with Preeclampsia

**DOI:** 10.3390/ijms25010122

**Published:** 2023-12-21

**Authors:** Arturo Flores-Pliego, Aurora Espejel-Nuñez, Hector Borboa-Olivares, Sandra Berenice Parra-Hernández, Araceli Montoya-Estrada, Humberto González-Márquez, Ramón González-Camarena, Guadalupe Estrada-Gutierrez

**Affiliations:** 1Department of Immunobiochemistry, Instituto Nacional de Perinatología, Mexico City 11000, Mexico or mxarturo.flores@inper.gob.mx (A.F.-P.); aurora.espejel@inper.gob.mx (A.E.-N.); sandrabpahdz@gmail.com (S.B.P.-H.); 2Postgraduate in Experimental Biology, Division of Biological and Health Sciences, Universidad Autónoma Metropolitana-Iztapalapa, Mexico City 09310, Mexico; 3Community Interventions Research Branch, Instituto Nacional de Perinatología, Mexico City 11000, Mexico; h_borboa1@yahoo.comhector.borboa@inper.gob.mx; 4Coordination of Gynecological and Perinatal Endocrinology, Instituto Nacional de Perinatología, Mexico City 11000, Mexico; ara_mones@hotmail.com; 5Health Science Department, Universidad Autónoma Metropolitana-Iztapalapa, Mexico City 09310, Mexico; humberto.gonzalez.marquez@gmail.com (H.G.-M.); rgc@xanum.uam.mx (R.G.-C.); 6Research Division, Instituto Nacional de Perinatología, Mexico City 11000, Mexico

**Keywords:** preeclampsia, endothelial dysfunction, MMP-2, IL-8

## Abstract

Endothelial dysfunction (ED) in preeclampsia (PE) results from the convergence of oxidative stress, inflammation, and alterations in extracellular matrix components, affecting vascular tone and permeability. The molecular network leading to ED includes IL-8 and MMP-2. In vitro, IL-8 regulates the concentration and activity of MMP-2 in the trophoblast; this interaction has not been studied in endothelial cells during PE. We isolated human umbilical vein endothelial cells (HUVECs) from women with healthy pregnancies (NP, *n* = 15) and PE (*n* = 15). We quantified the intracellular concentration of nitric oxide and reactive oxygen species with colorimetric assays, IL-8 with ELISA, and MMP-2 with zymography and using an ELISA-type system. An IL-8 inhibition assay was used to study the influence of this cytokine on MMP-2 concentration and activity. HUVECs from women with PE showed significantly higher oxidative stress than NP. IL-8 and MMP-2 were found to be significantly elevated in PE HUVECs compared to NP. Inhibition of IL-8 in HUVECs from women with PE significantly decreased the concentration of MMP-2. We demonstrate that IL-8 is involved in the mechanisms of MMP-2 expression in HUVECs from women with PE. Our findings provide new insights into the molecular mechanisms regulating the ED distinctive of PE.

## 1. Introduction

Hypertensive disorders during pregnancy remain a severe health issue worldwide. The adverse effects of elevated blood pressure range from mild to severe, leading to pre-eclampsia (PE) development, affecting practically all system organs, and contributing significantly to maternal and perinatal morbidity and mortality [1].

PE is characterized by defects in the placentation and inadequate trophoblast invasion of the spiral arteries to the uterine wall, causing placental ischemia/hypoxia as the first phase of this pathology. Endothelial dysfunction (ED) is a prominent feature during the second phase of PE, after 20th weeks of gestation, and it is considered a common event prior to the manifestation of clinical signs for PE diagnosis, such as hypertension (≥140/90 mmHg) and proteinuria (≥3 g/24 h) [2,3].

Women with PE exhibit decreased nitric oxide (NO) bioavailability and increased reactive oxygen species (ROS), which favor a pro-thrombotic and pro-inflammatory state that impairs the endothelium, with reduced vasodilation and increased vascular permeability, leading to generalized ED [4,5,6,7]. Sustained vascular release of pro-inflammatory cytokines, such as tumor necrosis factor-alpha (TNF-⍺), interleukin-1 (IL-1), IL-6, and IL-8, induce the chronic inflammatory state that is a feature of ED [8].

IL-8 is an inflammatory cytokine produced by different cell types such as macrophages, neutrophils, natural killer cells, and endothelial cells [9]. IL-8 stimulates the permeability of the damaged endothelium, promoting the progression of PE as it modulates the expression of molecules related to ED, causing endothelial junction damage, increased endothelial permeability, and synthesis of matrix metalloproteinases (MMPs) [10,11,12,13,14].

MMPs are a family of structurally related and highly regulated calcium- and zinc-dependent enzymes that degrade various extracellular matrix (ECM) components. Abnormal expression or an imbalance of different MMPs, i.e., MMP-1, MMP-2, MMP-9, and MMP-14, have been involved in the pathophysiology of PE [15,16,17,18]. MMP-2 degrades basement membrane collagens and is produced by many cell types, including endothelial cells [19,20].

In an in vitro ECM migration and invasion model using trophoblast HTR-8/SVneo cells, it was observed that IL-8 can induce the expression and activity of MMP-2 [21]. Moreover, IL-8 upregulates MMP-2 production and mRNA expression in endothelial cells in a model of angiogenesis [22]. The participation of IL-8 and MMP-2 in PE has been studied mainly in plasma and maternal serum, showing that both molecules are circulating at the systemic level, without delving into their origin and regulation. Therefore, this pilot study aimed to determine the effect of IL-8 on the concentration and activity of MMP-2 in HUVECs isolated from women with PE, as a possible mechanism implicated in the widespread ED observed in this pathology, and thus evaluate what could be occurring at the intracellular level.

## 2. Results

### 2.1. Clinical Characteristics of the Population

Clinical data of women included in the study are shown in Table 1. While no significant differences in maternal age, gestational weight gain, and platelet count between groups were found, other parameters, i.e., gestational age at birth, primiparous, systolic and diastolic blood pressure, protein/creatinine ratio, serum creatinine, glutamic oxaloacetic transaminase (GOT), and glutamic pyruvic transaminase (GPT) levels, were statistically different.

### 2.2. HUVECs Characterization

The endothelial origin of the isolated umbilical cord cells from women with PE and NP was demonstrated by positive staining for the vWf glycoprotein in the cytoplasm (Figure 1A), as well as for the cell adhesion molecule PECAM-1/CD31 at the intercellular junctions (Figure 1B).

### 2.3. Oxidative Stress and IL-8 Are Increased in HUVECs from Women with PE

HUVECs from women with PE showed a significant decrease in NO_2_^−^ and NO_3_^−^ concentration (*p* < 0.01) compared to cells from NP, as shown in Figure 2A. The opposite was observed for ROS generation, where a significant increase in the intensity of staining (*p* < 0.01) was observed in HUVECs from women with PE (Figure 2B,C). The quantification of IL-8 was performed in HUVEC lysates, observing a significantly higher concentration of this cytokine in PE (59.81 ± 11.29 pg/µg protein) vs. NP (27.69 ± 5.35 pg/µg protein) (*p* < 0.001) (Figure 3).

### 2.4. MMP-2 Is Increased in HUVECs from PE and IL-8 Regulates Its Concentration and Activity

Gelatin zymography showed lysis bands corresponding to proMMP-2 (72 kDa, inactive form) and MMP-2 (65 kDa, active form) in both study groups, being more intense in HUVECs from women with PE (Figure 4A). These results were corroborated in the ELISA-type assays, where concentrations of active MMP-2 and total MMP-2 (proenzyme and active form) were significantly increased in the group with PE (*p* < 0.001) (Figure 4B). Interestingly, the incubation of HUVECs from PE with a neutralizing anti-IL-8 antibody decreased the intensity of the lysis bands of MMP-2 and proMMP-2 in the gelatin zymography (Figure 5A), and significantly decreased the concentration of both forms of the enzyme (*p* < 0.001) in the quantitative ELISA-like assay (Figure 5B).

## 3. Discussion

In this research, we have found that HUVECs isolated from women with PE show increased signs of oxidative stress, characterized by a higher concentration of ROS and less NO metabolites compared to women with normal pregnancies, which is indicative of endothelial dysfunction. In addition to oxidative stress, an underlaying inflammatory process is also present in ED, and in this work we have described the increase in IL-8 and MMP-2, two modulators of inflammation, in the vascular endothelium of women with PE. Our findings also have revealed that IL-8 is implicated in the release and activity of MMP-2 in HUVECs, providing novel insights into the molecular mechanisms that regulate the extensive vascular endothelial damage observed in women with PE.

Endothelial dysfunction occurs during the second stage of PE as a consequence of the imbalance between different bioactive factors, including ROS and inflammatory cytokines that induce the expression of MMPs, which activate endothelial cells, damaging endothelial integrity [11,12,13,14]. Nitric oxide acts as an endothelial relaxant, and its effects as an antioxidant factor have been described in PE [23]. In this study, NO concentration, indirectly assessed through the quantification of its metabolites, was decreased in HUVEC lysates from women with PE. These results support what has been reported in the literature showing that during PE, high vascular resistance precedes ED, mainly influenced by decreased NO, which results in reduced vasodilation [24,25]. Furthermore, we also observed an imbalance between the decrease in NO, which acts as an antioxidant agent, and the increase in reactive oxygen species generation in cells from preeclamptic women, damaging cell binding proteins and leading HUVECs to ED. This imbalance may be caused by the ischemia and hypoxia events described during ED in PE. These findings are consistent with those previously reported by other authors.

The involvement of cytokines during PE has been a broad field of study. IL-8 is an inflammatory cytokine that can be produced locally due to tissue damage. The role of IL-8 secreted by endothelial cells during PE may be controversial, as results are inconsistent [26]. IL-8 can regulate chemotaxis, inflammation, and MMP production in an autocrine manner [14], and such events are relevant to ED; additionally, a previous work reports that endothelial cells are a source of IL-8 [27]. We found high levels of IL-8 in HUVECs from women with PE, consistent with other studies showing significantly higher concentrations of IL-8 in plasma from women with PE [28]. With this evidence, we can assume that IL-8 may have an active role in ED during PE, providing new insights into the molecular mechanisms involved in endothelial damage in women who develop this pathology.

Along with IL-8, we observed increased MMP-2 in HUVECs from women with pregnancies complicated with PE. Studies involving the analysis of MMP-2 during PE have mainly been performed in plasma, giving less importance to endothelial cells that are directly affected during PE and produce IL-8 and MMP-2. An important finding in this work was that although HUVECs from both study groups produce active MMP-2 (65 kDa), a higher concentration was found in HUVECs from PE. These high levels of MMP-2 may contribute to the degradation of collagen type I and III, one of the principal substrates of this enzyme [18,29] and identified in human umbilical cord stroma cells by other authors [30]. Thus, MMP-2 may exert its proteolytic activity by degrading collagen I and III and contribute to ED, leading to significant degradation of the basement membrane, possibly affecting the integrity of tight cell junction proteins such as claudins and cell adhesion molecules such as vascular endothelial cadherin (VE-CAM), causing permeability and endothelial damage [22]. Upon incubation of PE HUVECs with neutralizing anti-IL8 antibody, the concentration and activity of both forms of MMP-2 (proMMP-2 72 kDa and active MMP-2 65 kDa) decreased considerably. These findings allow us to propose that IL-8 exerts a considerable role in regulating the intracellular concentration of MMP-2 during ED, although the mechanism involved in this modulation has yet to be identified. In this regard, it has been described in a model of tuberculosis using epithelial cells that IL-17 can regulate MMP-3 expression, secretion, and activity, since this cytokine exerts a cooperative and synergistic effect by stabilizing the mRNA of other inflammatory cytokines that are related to the secretion of MMP-3 and its possible activators [31].

The strengths of our work include that the HUVECs used were isolated directly from women with PE as a natural in vitro model of the disease, and the endothelial origin was ensured by the presence of two highly selective markers for this cell population: the vWf in the cytoplasm [32,33] and CD31 at the intercellular junctions [34]. On the other hand, women in both study groups had an adequate gestational weight gain, because it has been demonstrated that an excessive weight gain during pregnancy leads to ED [35]. Moreover, only women with pregnancies interrupted by cesarean section were included to prevent the onset of inflammatory mechanisms characteristic of labor that would confound our findings [36,37]. Limitations of our study include that a cell culture does not allow us to evaluate all the interactions that occur in vivo, where endothelial cells connect with other cell linages and different molecular mediators. We also recognize that HUVECs do not fully represent the complexity and heterogeneity of ED during preeclampsia.

The results of this pilot study are the basis for our future research, which will be focused on determining the IL-8-induced intracellular signaling pathways and transcription factors that bind to MMP-2 promoter in vascular endothelial cells of women with pre-eclampsia

## 4. Materials and Methods

### 4.1. Ethics Statement

This study was approved by the Institutional Review Board of the Instituto Nacional de Perinatologia in Mexico City (212250-3210-21201-02-16). Participants signed an informed consent form before enrollment.

### 4.2. Study Groups

Thirty pregnant women were included in the study: fifteen women with healthy pregnancies (normal pregnancy = NP) and fifteen with PE. Participants in both groups had a singleton delivery by cesarean section, without clinical evidence of labor. Exclusion criteria were clinical obesity data; gestational or pre-gestational diabetes; intrauterine infection; and renal, immunological, or cardiovascular diseases.

PE was defined using the American College of Obstetricians and Gynecologists (ACOG) criteria as systolic blood pressure ≥ 140 mm of mercury (mmHg) or a diastolic blood pressure ≥ 90 mmHg on two or more occasions at least 4 h apart after 20 weeks of gestation in women with previously normal blood pressure, and proteinuria ≥ 300 mg/24 h) [3]. The NP group included women who delivered at term (>37 weeks of gestation) and showed no postpartum evidence of hypertensive disease.

### 4.3. HUVECs Isolation

HUVECs were isolated from pre-eclamptic or normal pregnant women, using a previously described method [38]. Briefly, umbilical cord veins were infused with collagenase type I (250 U/mL) (Cat. 17100017; Thermo Fisher Scientific, MA, USA) for 15 min at 37 °C and lightly massaged to dislodge the cells. The eluted volume after digestion was collected and centrifuged at 1200 rpm for 5 min to obtain the cell package. The viability and cell number were determined using a Neubauer counting chamber by exclusion with 0.4% *w*/*v* trypan blue (Cat. 72-57-1; Sigma-Aldrich, St. Louis, MO, USA).

### 4.4. HUVEC Identification by Immunofluorescence

HUVECs were cultured for 24 h in 4-well Lab-Tek cell culture chamber slides (Cat. 154461PK; Nalgene Nunc; Naperville, IL, USA) with supplemented EndoGRO-LS medium (Cat. SCME001; Millipore, Burlington, MA, USA) at a density of 5 × 10^4^ cells per well. Cells were washed with 1X PBS (Cat. 10010015; Gibco, Grand Island, NY, USA), fixed with 3.7% paraformaldehyde/PBS 1X, and permeabilized with 2% Triton X-100I do not confirm (Cat 9036-19-5; Sigma-Aldrich, St. Louis, MO, USA). Cells were then blocked with human IgG for 20 min at room temperature. Endothelial cells were incubated overnight at 4 °C in a humid chamber with anti-von Willebrand factor (vWf) antibodies conjugated to the fluorescent dye Alexa Fluor^®^ 488 (FITC) (Cat. 195028; Abcam, Cambridge, UK; dilution 1:100) and anti-CD31 endothelial and platelet cell adhesion molecule (PECAM-1) coupled with Alexa Fluor 647 (Cat. Ab218582; Abcam, MA, USA; diliution 1:500) [39]. Cell nuclei were evidenced via DNA staining when mounted in a VectaShield vibrance antifade mounting medium containing 4′,6-diamidino-2-phenylindole (DAPI) (Cat. HG-1800; Vector, CA, USA). Cells were then analyzed using a laser scanning microscope Axiovert 200 M (Carl Zeiss, Oberkochen, Germany).

### 4.5. Cell Culture

HUVECs were cultured in EndoGRO-LS medium supplemented with 5 ng/mL human recombinant EGF (epidermal growth factor), 50 µg/mL ascorbic acid, 10 mM L-glutamine, 1 µg/mL hydrocortisone hemisuccinate, 0.75 U/mL heparan sulfate, and 2% fetal bovine serum. Cell cultures were grown in 25 cm^2^ cell culture bottles at a density of 2.8 × 10^6^ and maintained at 5% CO_2_, 37 °C, and 95% humidity. The HUVECs were used only between the first and third passes for protein extraction.

### 4.6. Protein Extraction

Once confluent, HUVECs cultures were washed twice with 1X PBS and protein extraction was performed using the commercial cell lysis reagent M-PER (Cat. 78501; Thermo Fisher Scientific, Waltham, MA, USA), according to the manufacturer’s instructions. A Halt^®^ protease and phosphatase inhibitor (Cat. 78445; Thermo Fisher Scientific, MA, USA) was added at a concentration of 1X. Cell lysis was performed at 4 °C, and cell debris was removed by centrifugation at 14,000× rpm for 10 min at 4 °C. Protein concentration was determined using the bicinchoninic acid (BCA) method (Cat. 23227; Thermo Fisher Scientific, MA, USA) on the microplate [40]. Cell lysates were stored at −70 °C until processing.

### 4.7. Quantification of NO in HUVECs

NO concentrations were determined indirectly by the quantification of their end products, nitrite (NO_2_^−^) and nitrate (NO_3_^−^), using a colorimetric assay (Cat. 780001; Cayman Chemical Company, Ann Arbor, MI, USA), according to the manufacturer’s instructions. Briefly, cell lysates from both study groups were ultra-filtrated using a molecular cutoff of 10,000 to 6000× *g* for 60 min at 4 °C. The ultra-filtrated cell lysates were incubated with nitrate reductase and its cofactor; then, the Griess reagent was added and incubated for 15 min. Samples were measured at 540 nm using a microplate reader (Synergy HT, Biotek).

### 4.8. Intracellular ROS

Intracellular ROS production by HUVECs of NP and PE was determined using a 2′, 7′-dichlorodihydrofluorescein diacetate reagent (DCFH-DA) (Cat. D399; Invitrogen, Carlsbad, CA, USA). HUVECs from both study groups were cultured in 4-well chamber slides for 24 h, washed with 1X PBS, and incubated with EndoGRO-LS culture medium containing DCFH-DA 10 µmol/L for 15 min at 37 °C in the dark. Cells were washed twice with 1X PBS, fixed with methanol, and preserved with a DAPI-conjugated fluorescence mounting medium, VectaShield vibrance antifade (Cat. HG-1800; Vector, CA, USA). Samples were analyzed on a laser scanning microscope Axiovert 200 M LSM 510 Meta (Carl Zeiss, Oberkochen, Germany), and the signal was acquired at 488 nm. The Fiji-ImageJ software 1.53f (http://imagej.nih.gov/ij accesed on 13 August 2023 National Institute of Mental Health, Bethesda, MD, USA) was used to obtain the fluorescence intensity values [41].

### 4.9. IL-8 Quantification

IL-8 concentration in lysates of HUVECs from both study groups was determined with the Quantikine IL-8/CXCL8 human ELISA kit (Cat. D8000C; R&D Systems, Minneapolis, MN, USA) with a range of 31.2–2000 pg/mL and sensitivity of 7.5 pg/mL. Sample readings were obtained using a microplate reader (Synergy HT, Biotek Instruments, Winooski, VT, USA) at 450 nm according to the manufacturer’s instructions.

### 4.10. Gelatin Zymography for MMP-2

Gelatinase activity was determined in HUVEC lysates of NP and PE via sodium dodecyl sulfate-polyacrylamide gel electrophoresis (SDS-PAGE) containing 8% polyacrylamide copolymerized with 1% porcine skin type A gelatin (cat. G2625; Sigma-Aldrich, MO, USA) under non-denaturing conditions, with a mini gel-formatted system as described above [42]. Activity markers for MMP-2 and MMP-9 were included in each gel using the supernatant obtained from the U937 promyelocyte cell line. Zones of enzyme activity appeared as clear lysis bands against a background of the non-degraded substrate.

### 4.11. MMP-2 Quantification

Simultaneously to zymography, the concentration of proMMP-2 and the MMP-2 active (MMP-2 total) form was determined in HUVECs lysates of NP and PE using an ELISA-type system (Cat. RPN 2617; Biotrak™ Amersham Biosciences; Buckinghamshire, UK), which incorporates a capture system with monoclonal antibodies and the use of a specific substrate for MMP-2, with a specificity of 190 pg/mL–0.5 ng/mL.

### 4.12. Inhibition of IL-8

The influence of IL-8 on the concentration and activity of MMP-2 was determined in cultures of HUVECs from women with PE at 80% confluence that were incubated with neutralizing anti-IL-8 antibody (Cat. MBA208; R&D Systems, MN, USA) 0.4 µg/mL for 24 h, under the previously described culture conditions. At the end of the incubation time, the cells were washed with 1X PBS to remove any medium remanent, and HUVEC lysates were then analyzed using zymography and the ELISA-type system for MMP-2, as described above.

### 4.13. Statistical Analysis

Analysis was performed using Student’s *t*-test or Mann–Whitney *U* test for parametric and nonparametric data distribution, respectively, with a significance level of *p* ≤ 0.05. The results obtained were expressed as mean ± standard deviation (SD). Statistical analysis was performed with GraphPad Prism 8 software for Mac.

## 5. Conclusions

In conclusion, this study shows that HUVECs isolated from women with PE secrete elevated concentrations of IL-8, and this cytokine may act as a regulatory molecule of MMP-2 concentration and activity as a possible mechanism implicated in the extensive ED observed in women with PE.

## Figures and Tables

**Figure 1 ijms-25-00122-f001:**
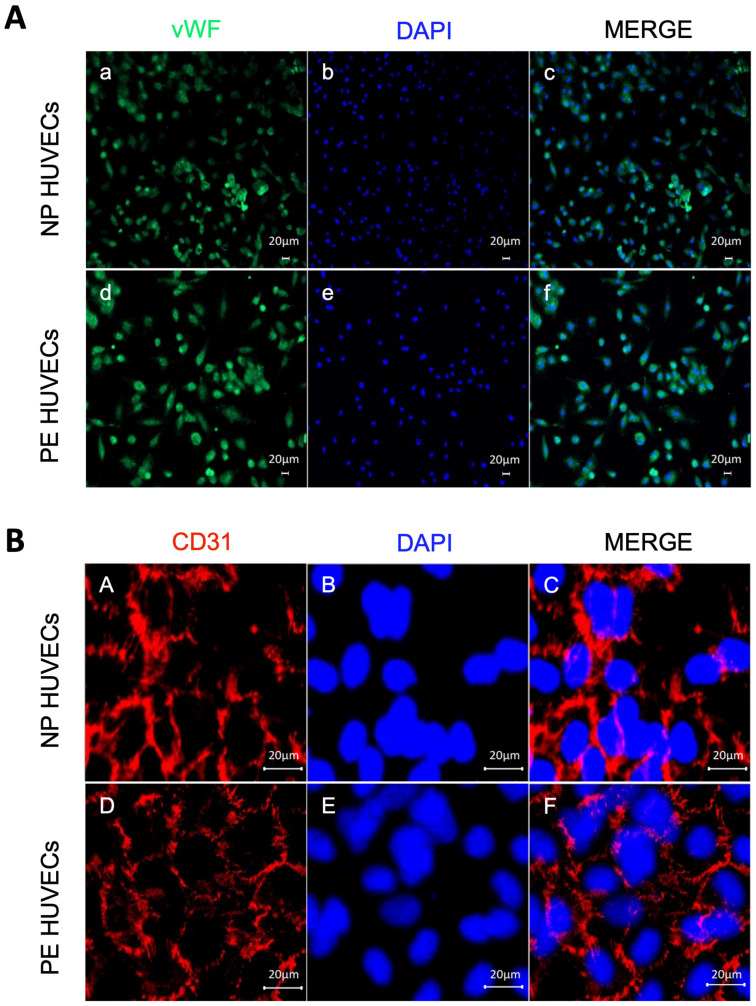
Endothelial origin of the isolated umbilical vein cells (HUVECs). Monolayer staining of HUVECs isolated from both normal pregnancy (NP) or PE (preeclampsia) study groups showing (**A**) (**a**,**d**) nuclear and cytoplasmic immunofluorescence for von Willebrand factor, (**b**,**e**) DAPI nuclei staining, and (**c**,**f**) Merge of von Willebrand factor and DAPI. Representative images of three different experiments for each study group. Magnification 10×. Scale bar = 20 µm; (**B**) (**A**,**D**) intercellular junction immunofluorescence for CD31 adhesion molecule, (**B**,**E**) DAPI nuclei staining, and (**C**,**F**) merge of CD31 and DAPI. Representative images of three different experiments for each study group. Magnification 20×; zoom 2; scale bar = 20 µm.

**Figure 2 ijms-25-00122-f002:**
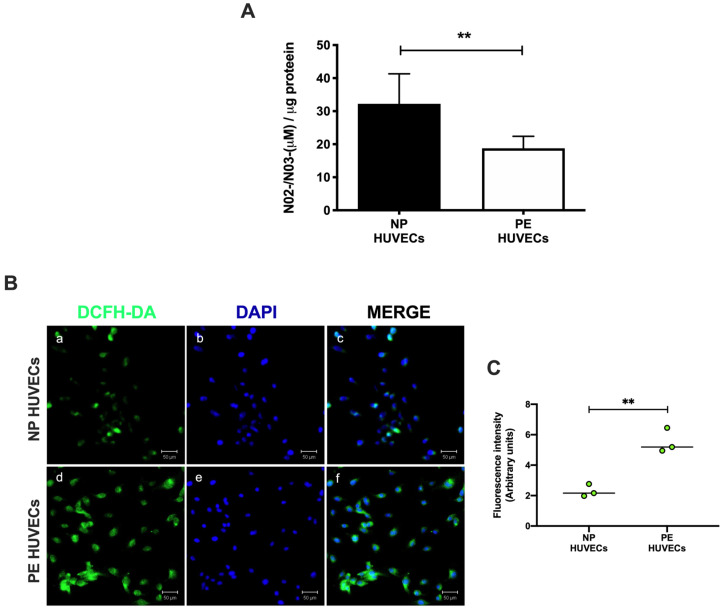
Oxidative stress is increased in umbilical vein cells (HUVECs) isolated from women with preeclampsia. (**A**) Nitrite and nitrate as stable nitric oxide metabolites in HUVEC lysates from normal pregnancy (NP) or preeclampsia (PE). Data are expressed as the mean ± SD of 15 experiments in duplicate. ** *p* < 0.01. (**B**) Intracellular reactive oxygen species (ROS) production in HUVEC monolayers from NP and PE. (**a**,**d**) Immunofluorescence for ROS with the 2′,7′-dichlorodihydrofluorescein diacetate reagent (DCFH-DA), (**b**,**e**) DAPI nuclei staining, and (**c**,**f**) merge of ROS and DAPI. Representative images of three different experiments for each study group. Magnification 20×. Scale bar = 50 µm. (**C**) Quantification of mean fluorescence intensity in three independent images from each condition. Student’s *t*-test. ** *p* < 0.01. The image has the correct resolution according to the authors’ guide.

**Figure 3 ijms-25-00122-f003:**
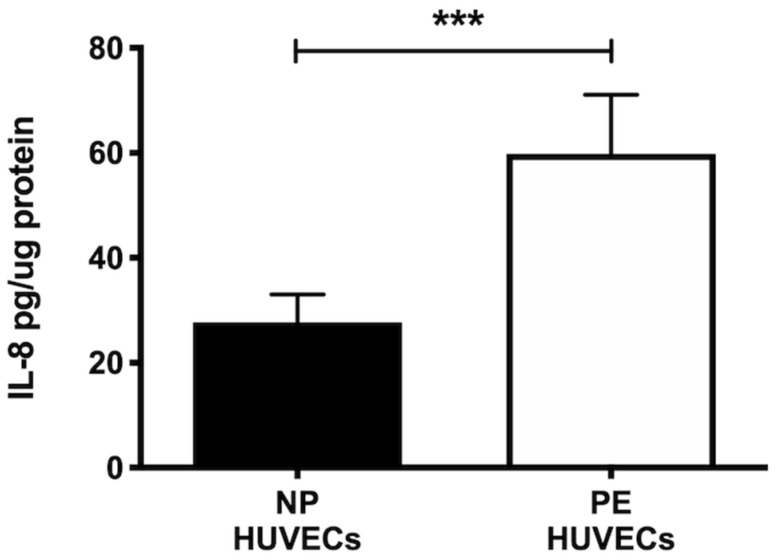
IL-8 is increased in umbilical vein cells (HUVECs) isolated from women with preeclampsia. Quantification of IL-8 using ELISA in cell lysates of HUVECs from women with normal pregnancy (NP) or preeclampsia (PE). Data are expressed as mean ± SD of 15 experiments in duplicate, *** *p* < 0.001.

**Figure 4 ijms-25-00122-f004:**
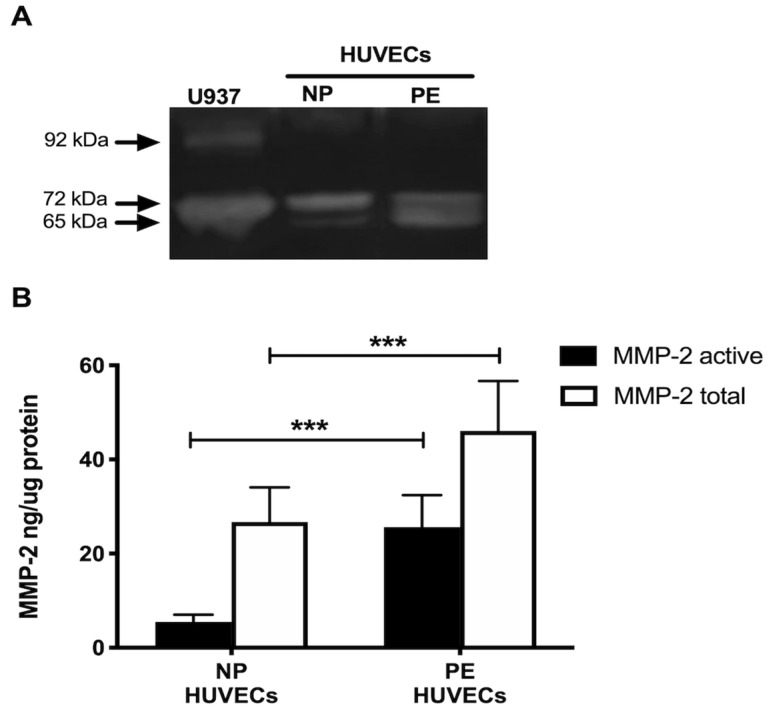
MMP-2 is increased in umbilical vein cells (HUVECs) isolated from women with preeclampsia. (**A**) Representative gelatin zymography (0.5 μg of protein per lane) of HUVEC lysates from women with normal pregnancy (NP) or preeclampsia (PE), showing lysis bands corresponding to proMMP-2 (72 kDa, inactive form) and MMP-2 (65 kDa, active form). (**B**) Quantification of MMP-2 enzymatic activity using ELISA-type assay in HUVEC lysates from both study groups. Bars represent the mean ± SD of 15 experiments in duplicate, *** *p* < 0.001.

**Figure 5 ijms-25-00122-f005:**
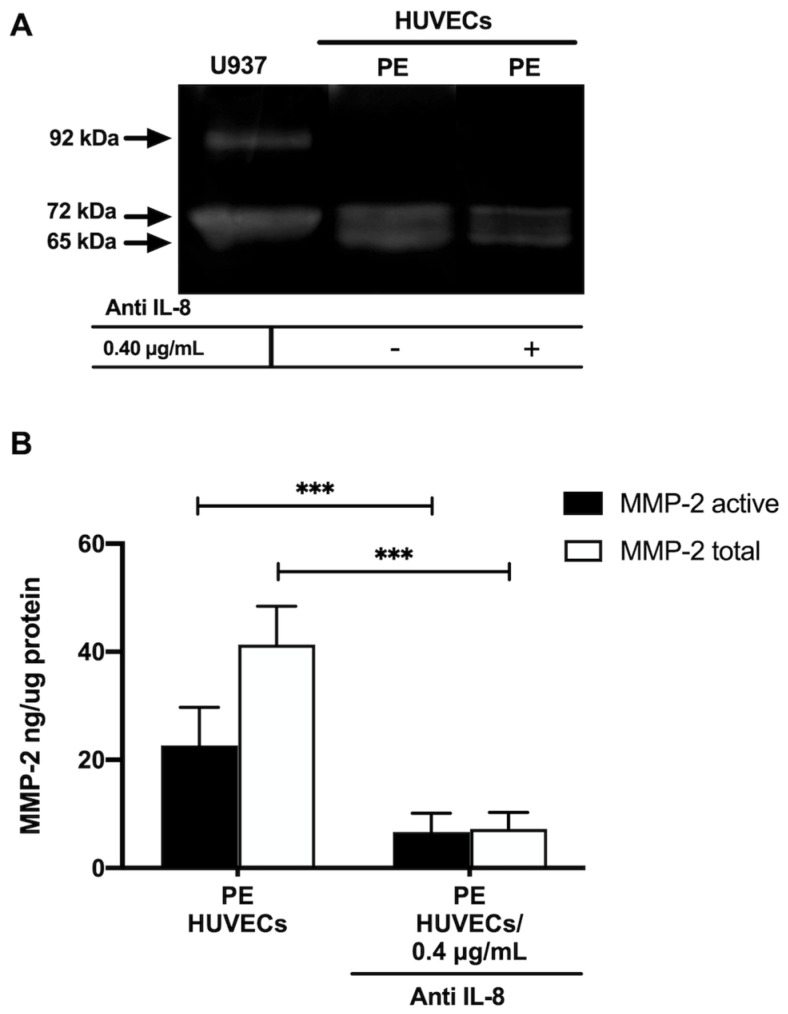
IL–8 regulates the concentration and activity of MMP-2 in umbilical vein cells (HUVECs) isolated from women with preeclampsia. (**A**) Representative gelatin zymography (0.5 μg of protein per lane) of HUVEC from women with preeclampsia (PE) incubated with a neutralizing anti-IL–8 antibody, showing lysis bands corresponding to proMMP-2 (72 kDa, inactive form) and MMP-2 (65 kDa, active form). (**B**) Quantification of MMP-2 enzymatic activity using ELISA-type assay in HUVEC from women with PE incubated with a neutralizing anti-IL–8 antibody. Bars represent the mean ± SD of 15 experiments in duplicate, *** *p* < 0.001.

**Table 1 ijms-25-00122-t001:** Clinical data of women included in the study.

	NP (*n* = 15)	PE (*n* = 15)
Maternal Age (years)	29.3 ± 7.6	31.2 ± 8.3
Gestational age at birth (weeks)	38.3 ± 3.4	33 ± 1.5 *
Gestational weight gain (kg)	9.1 ± 2.3	7.3 ± 1.3
Primiparous (%)	0	16 *
Systolic blood pressure (mmHg)	111.4 ± 7.6	144.6 ± 4.1 *
Diastolic blood pressure (mmHg)	74.7 ± 5.7	92.8 ± 7.0 *
Protein/creatinine ratio (mg/dL)	0.14 ± 0.1	0.31 ± 0.1 *
Creatinine (mg/dL)	0.52 ± 0.02	0.65 ± 0.03 *
GOT (uI/L)	13.63 ± 3.6	21 ± 5.5 *
GPT (uI/L)	12.88 ± 1.3	24.13 ± 9.8 *
Platelet count (×10^9^ L)	274.2 ± 10.5	243.2 ± 10.3

Data presented as mean ± standard deviation, unless otherwise indicated. Mann–Whitney U test. * *p* ≤ 0.05. GOT, glutamic oxaloacetic transaminase; GPT, glutamic pyruvic transaminase.

## Data Availability

The data sets used and/or analyzed during the current study are available from the corresponding author upon reasonable request.

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
