# Peer review of "Regulation of MMP-2 by IL-8 in Vascular Endothelial Cells: Probable Mechanism for Endothelial Dysfunction in Women with Preeclampsia"

_ijms, 2023, doi:10.3390/ijms25010122_

Round 1

Reviewer 1 Report

Comments and Suggestions for Authors

Authors present interesting findings on the effects of PE on isolated umbilical vein endothelial cells. I have a few major comments due to some issues and weaknesses in the paper.

for figure 2: DCFH image for PE shows nonspecific staining that is strong but not even on the cells themselves. This is not acceptable for quantification and artificially increases fluorescence. Authors should be doing this in a plate reader anyway with far more replicate wells in a black plate for an unbiased quantification.

eNOS and eNOS phosphorylation should also be assessed. There is a disconnect with the NO data.

Author Response

Thank you for your comments on our manuscript " Regulation of MMP-2 by IL-8 in Vascular Endothelial Cells: Probable Mechanism for Endothelial Dysfunction in Women with Preeclampsia." (ijms-2736917).We have addressed all your comments, which contributed to making our manuscript significantly better.

Reviewer. - Authors present interesting findings on the effects of PE on isolated umbilical vein endothelial cells. I have a few major comments due to some issues and weaknesses in the paper.

For figure 2: image for PE shows nonspecific staining that is strong but not even on the cells themselves. This is not acceptable for quantification and artificially increases fluorescence. Authors should be doing this in a plate reader anyway with far more replicate wells in a black plate for an unbiased quantification.

Answer: The signal intensity observed in our immunofluorescence images for DFCH-DA in PE samples were obtained with the same laser excitation-intensity parameters as the control samples (active line 488 nm, at a transmission 14% power and a scan speed of 4), so the higher signal intensity in PE HUVECs is because they have a higher positivity to DFCH-DA. Moreover, in case of autofluorescence or un-specificity in the staining emitted from endogenous fluorescent molecules, the signals are weak even when the molecules are irradiated with high power excitation light (doi: 10. 1002/bies.201700003) (doi.org/10.3389/fbioe.2021.789709).

In the confocal microscopy for DFCH-DA, the intensity of the staining could not be nonspecific because if we had used a very high laser power to observe the sample, this intense and prolonged excitation would lead to a degradation of the dye and, therefore, to a decrease of the signal. Moreover, we know that the staining for DFCH-DA (green) is specific and is in the cell cytoplasm because the polygonal morphology characteristic of HUVECs is clearly observed in green, while the nuclei are stained blue (DAPI).

We appreciate your suggestion to perform the assay to quantify DFCH-DA in microplate to obtain quantitative results; however, it is unusual to quantify immunostaining with the proposed method (reading on a black plate), and we consider that it is acceptable to evaluate inmmunofluorescence with the method used in our work, as it has been reported in most papers.

Reviewer. - eNOS and eNOS phosphorylation should also be assessed. There is a disconnect with the NO data.

Answer. - We understand the importance of eNOS, its role in NO synthesis, and that phosphorylation processes regulate its function. However, it has been described that NO assessment alone is considered a central marker in the context of endothelial dysfunction in PE, as changes in NO concentration may lead to the onset of organ failure through inflammatory processes (doi: 10.3390/ijms16034600). Additionally, we would like to point out that the main focus of our study was specifically on the regulation of MMP-2 by IL-8 in endothelial cells.

Dr. Guadalupe Estrada-Gutierrez

Reviewer 2 Report

Comments and Suggestions for Authors

In this study, the group of Estrada-Gutierrez and colleagues aim to strengthen previously published findings on vascular endothelial cell dysfunction in preeclampsia (PE). Authors confirmed that HUVECs from women with PE showed increased signs of oxidative stress, higher levels of ROS measured by DCFH-DA and less NO metabolites than women with healthy pregnancies (NP.). They also found elevated concentrations of IL-8 and MMP-2 in PE HUVECs compared to NP. More importantly, attempts were also made to find a role of IL-8 in regulating MMP-2. To this end, IL-8 inhibition assay was implied; inhibition of IL-8 in PE HUVECs decreased the concentration and activity of MMP-2. The overall impression is that the manuscript describes some preliminary results related to the mechanism regulating MMP-2 rather than a well-designed experiments.

The relation of oxidative stress to endothelial dysfunction in preeclampsia has been known for years, see for instance doi.org/10.1111/j.1447-0756.2009.01128.x. Same for increased IL-8 levels (doi.org/10.1016/S0029-7844(02)02169-5, doi.org/10.1111/j.1600-0897.2007.00486.x, doi.org/10.3109/14767051003774942) and MMP2 (doi.org/10.1081/PRG-120028281) in PE women. These parameters are well known candidate for markers of endothelial dysfunction in PE. Therefore, I think that most of the results of the present work only confirm previous findings, and the paper add very little new knowledge. Although the work done is important, the biological sound of the paper is rather weak in its present form. In my opinion, this work requires more experiments, e.g. some functional analyses, more insight into the molecular mechanism of MMP-2 regulation by IL-8 etc. to make it more valuable and interesting.

Other comments:

Abstract

·        L22: „in vitro” Capital letter

·        L27-28: „An IL-8 inhibition assay determined the role of IL-8 on the concentration and activity of MMP-2 HUVECs”. This sentence is redundant.

·        L31: „of both forms of MMP-2”, specify these forms or do not mention „forms” here

Introduction

·        Li et al., 2003 showed that IL-8 upregulates MMP-2 production and mRNA expression in HUVEC. Therefore, this paper, Ref [30] should be cited in the introduction to support authors’ hypothesis and goals.

Results

·        Table 1. Please specify where Student’s t-test or Mann Whitney U test were used.

·        Figure 1A: Positive staining for the vWf glycoprotein is also visible in the nucleus. Results section and figure legend should be correced.

·        Figure 1 B: poor quality of images; too much color intensity, probably too long exposure time 

·        For immunofluorescence, I would like to see a negative control.

·        Figure 2. In the image I see scale bar = 50 μm and in the legend 20 μm. Please check and correct it.

Author Response

Thank you for your comments on our manuscript " Regulation of MMP-2 by IL-8 in Vascular Endothelial Cells: Probable Mechanism for Endothelial Dysfunction in Women with Preeclampsia." (ijms-2736917).We have addressed all your comments, which contributed to making our manuscript significantly better.

In this study, the group of Estrada-Gutierrez and colleagues aim to strengthen previously published findings on vascular endothelial cell dysfunction in preeclampsia (PE). Authors confirmed that HUVECs from women with PE showed increased signs of oxidative stress, higher levels of ROS measured by DCFH-DA and less NO metabolites than women with healthy pregnancies (NP.). They also found elevated concentrations of IL-8 and MMP-2 in PE HUVECs compared to NP. More importantly, attempts were also made to find a role of IL-8 in regulating MMP-2. To this end, IL-8 inhibition assay was implied; inhibition of IL-8 in PE HUVECs decreased the concentration and activity of MMP-2. The overall impression is that the manuscript describes some preliminary results related to the mechanism regulating MMP-2 rather than a well-designed experiments.

Reviewer. - The relation of oxidative stress to endothelial dysfunction in preeclampsia has been known for years, see for instance doi.org/10.1111/j.1447-0756.2009.01128.x. Same for increased IL-8 levels (doi.org/10.1016/S0029-7844(02)02169-5, doi.org/10.1111/j.1600-0897.2007.00486.x, doi.org/10.3109/14767051003774942) and MMP2 (doi.org/10.1081/PRG-120028281) in PE women. These parameters are well known candidate for markers of endothelial dysfunction in PE. Therefore, I think that most of the results of the present work only confirm previous findings, and the paper add very little new knowledge. Although the work done is important, the biological sound of the paper is rather weak in its present form. In my opinion, this work requires more experiments, e.g., some functional analyses, more insight into the molecular mechanism of MMP-2 regulation by IL-8 etc. to make it more valuable and interesting.

Answer. – As reviewer points out, endothelial dysfunction in preeclampsia has been previously described as well as the increase in cytokines such as IL-8, and enzymes like MMP-2. However, it is important to highlight that most determinations of NO, ROS, IL-8, and MMP-2 in preeclampsia have been performed in maternal plasma or serum, showing what happens systemically during preeclampsia. In contrast, in our work, we analyzed the intracellular concentrations of these molecules in HUVECs of women with preeclampsia and healthy pregnancies to explore the local contribution of these cells to the dysfunction suffered by the endothelium during preeclampsia, further showing that IL-8 is involved in the regulation of MMP-2 concentration in these cells.

In our study we performed a functional assay by using an antibody to block IL-8, in which the concentration of MMP-2 was shown to decrease. As suggested by the reviewer, the next step will be to perform further experiments to learn the molecular mechanism involved, which we hope to report in a future study, as this is a priority line of research in our laboratory.

Other comments:

Abstract

Reviewer. - L22: “in vitro” Capital letter.

Answer. – This correction has already been addressed.

Reviewer. - L27-28: “An IL-8 inhibition assay determined the role of IL-8 on the concentration and activity of MMP-2 HUVECs”. This sentence is redundant.

Answer. - The sentence was changed to “An IL-8 inhibition assay was used to study the influence of this cytokine on MMP-2 concentration and activity.” Thank you for the observation.

Reviewer. - L31: “of both forms of MMP-2”, specify these forms or do not mention “forms” here.

Answer. - We have corrected the statement by not mentioning the forms of MMP-2 as “decreased the concentration of MMP-2”.

Introduction

Reviewer. - Li et al., 2003 showed that IL-8 upregulates MMP-2 production and mRNA expression in HUVEC. Therefore, this paper, Ref [30] should be cited in the introduction to support authors’ hypothesis and goals.

Answer. - We have included in the introduction the following statement: “Moreover, IL-8 upregulates MMP-2 production and mRNA expression in endothelial cells in a model of angiogenesis” (Li et al., 2003). Lanes 66-67.

Results

Reviewer. - Table 1. Please specify where Student’s t-test or Mann Whitney U test were used.

Answer. - We have corrected this statement in Table 1 indicating that we used the Mann Whitney U test (due to the number of measurements in each group). Now is read as: “Data presented as mean ± standard deviation, unless otherwise indicated. Mann-Whitney U test. *p≤0.05. GOT, Glutamic oxaloacetic transaminase; GPT, glutamic pyruvic transaminase”.

Reviewer. - Figure 1A: Positive staining for the vWf glycoprotein is also visible in the nucleus. Results section and figure legend should be corrected.

Answer. - We have corrected now is read "A) (a,d) Nuclear and cytoplasmic immunofluorescence for von Willebrand factor".

Reviewer. - Figure 1 B: poor quality of images; too much color intensity, probably too long exposure time.

Answer. – The image has been corrected.

Reviewer. - For immunofluorescence, I would like to see a negative control.

Answer. – Unfortunately, we do not have the images of the negative control in the immunofluorescence. At the moment it is impossible to obtain the requested images due to some technical failures of our confocal microscope.

Reviewer. - Figure 2. In the image I see scale bar = 50 μm and in the legend 20 μm. Please check and correct it.

Answer. – This scale bar data has been revised and corrected in the caption of figure 2. "Magnification 20X. Scale bar = 50 µm".

Dr. Guadalupe Estrada-Gutierrez

Reviewer 3 Report

Comments and Suggestions for Authors

Title: Regulation of MMP-2 by IL-8 in Vascular Endothelial Cells: Probable Mechanism for Endothelial Dysfunction in Women with Preeclampsia.

Authors: Arturo Flores-Pliego, Aurora Espejel-Nuñez, Hector Borboa-Olivares, Sandra Berenice Parra-Hernández, Araceli Montoya-Estrada, Humberto González-Márquez, Ramón González-Camarena, Guadalupe Estrada-Gutierrez

Comments for the authors

The authors present the regulation of MMP-2 and IL-8 in a HUVECs model derived from pregnancies complicated by preeclampsia. The literature is scarce in this field and the topic is novum. I would like to recommend the manuscript for publication with minor remarks. I think that it is an interesting topic for the public of International Journal of Molecular Sciences. It is a well-written article.

The authors provide a succinct literature overview of endothelial dysfunction in PE and they provide that IL-8 and MMP-2 is increased in the endothelial cells of umbilical vein as markers of a higher oxidative stress.

I have some minor remarks:

First, the authors mixed up the order of the sections:

Section nr 2 is the Results which should be section nr 4 and the Discussion section should be nr 5, while Section nr 2 should be the Materials and Methods.

In the abstract section:

Typographic error in line 22: 'in vitro…' the sentence is not capitalized.

Line 28: MMP-2 in HUVECs.

Line 75 GOT and GPT 'levels'

Line 248-249: the authors should exclude all cardiovascular disease as well.

Author Response

Thank you for your comments on our manuscript " Regulation of MMP-2 by IL-8 in Vascular Endothelial Cells: Probable Mechanism for Endothelial Dysfunction in Women with Preeclampsia." (ijms-2736917).We have addressed all your comments, which contributed to making our manuscript significantly better.

The authors present the regulation of MMP-2 and IL-8 in a HUVECs model derived from pregnancies complicated by preeclampsia. The literature is scarce in this field and the topic is novum. I would like to recommend the manuscript for publication with minor remarks. I think that it is an interesting topic for the public of International Journal of Molecular Sciences. It is a well-written article.

The authors provide a succinct literature overview of endothelial dysfunction in PE and they provide that IL-8 and MMP-2 is increased in the endothelial cells of umbilical vein as markers of a higher oxidative stress.

I have some minor remarks:

Reviewer. - First, the authors mixed up the order of the sections: Section nr 2 is the Results which should be section nr 4 and the Discussion section should be nr 5, while Section nr 2 should be the Materials and Methods.

Answer. – Please note that the order of the sections that appear in our manuscript is according to the IJMS instructions for authors: Introduction, Results, Discussion, Materials and Methods, and Conclusions.

In the abstract section:

Reviewer. - Typographic error in line 22: 'in vitro…' the sentence is not capitalized.

Answer. – We have corrected and capitalized "in vitro" to "In vitro."

Reviewer. - Line 28: MMP-2 in HUVECs.

Answer. – This statement has been changed to “An IL-8 inhibition assay was used to study the influence of this cytokine on MMP-2 concentration and activity.

Reviewer. - Line 75 GOT and GPT 'levels'

Answer. –The word “levels” was added (Line 77)

Reviewer. - Line 248-249: the authors should exclude all cardiovascular disease as well.

Answer. – The word cardiomyopathy was eliminated and changed to cardiovascular diseases, which is the population to which we really wanted to refer.  Now is read as "Exclusion criteria were clinical obesity data, gestational or pre-gestational diabetes, intrauterine infection, and renal, immunological, or cardiovascular diseases."

Dr. Guadalupe Estrada-Gutierrez

Round 2

Reviewer 1 Report

Comments and Suggestions for Authors

No further comments.

Author Response

Dear Reviewer.

Thank you for your comments on our manuscript " Regulation of MMP-2 by IL-8 in Vascular Endothelial Cells: Probable Mechanism for Endothelial Dysfunction in Women with Preeclampsia" (ijms-2736917), which contributed to making our manuscript significantly better.

Dr. Guadalupe Estrada-Gutierrez
